# Deep Learning Convolutional Neural Network Reconstruction and Radial k-Space Acquisition MR Technique for Enhanced Detection of Retropatellar Cartilage Lesions of the Knee Joint

**DOI:** 10.3390/diagnostics13142438

**Published:** 2023-07-21

**Authors:** Malwina Kaniewska, Eva Deininger-Czermak, Maelene Lohezic, Falko Ensle, Roman Guggenberger

**Affiliations:** 1Institute of Diagnostic and Interventional Radiology, University Hospital Zurich (USZ), Raemistrasse 100, 8091 Zurich, Switzerland; eva.deininger@usz.ch (E.D.-C.); falko.ensle@usz.ch (F.E.); roman.guggenberger@usz.ch (R.G.); 2Institute of Diagnostic and Interventional Radiology, University of Zurich (UZH), Raemistrasse 100, 8091 Zurich, Switzerland; 3Department of Forensic Medicine and Imaging, Institute of Forensic Medicine, University of Zurich, 8152 Zurich, Switzerland; 4Advanced Technology, Science and Technology Organization, GE HealthCare, 8152 Zurich, Switzerland; maelene.lohezic@ge.com

**Keywords:** MRI, PROPELLER technique, artificial intelligence, knee

## Abstract

Objectives: To assess diagnostic performance of standard radial k-space (PROPELLER) MRI sequences and compare with accelerated acquisitions combined with a deep learning-based convolutional neural network (DL-CNN) reconstruction for evaluation of the knee joint. Methods: Thirty-five patients undergoing MR imaging of the knee at 1.5 T were prospectively included. Two readers evaluated image quality and diagnostic confidence of standard and DL-CNN accelerated PROPELLER MR sequences using a four-point Likert scale. Pathological findings of bone, cartilage, cruciate and collateral ligaments, menisci, and joint space were analyzed. Inter-reader agreement (IRA) for image quality and diagnostic confidence was assessed using intraclass coefficients (ICC). Cohen’s Kappa method was used for evaluation of IRA and consensus between sequences in assessing different structures. In addition, image quality was quantitatively evaluated by signal-to-noise ratio (SNR) and contrast-to-noise ratio (CNR) measurements. Results: Mean acquisition time of standard vs. DL-CNN sequences was 10 min 3 s vs. 4 min 45 s. DL-CNN sequences showed significantly superior image quality and diagnostic confidence compared to standard MR sequences. There was moderate and good IRA for assessment of image quality in standard and DL-CNN sequences with ICC of 0.524 and 0.830, respectively. Pathological findings of the knee joint could be equally well detected in both sequences (κ-value of 0.8). Retropatellar cartilage could be significantly better assessed on DL-CNN sequences. SNR and CNR was significantly higher for DL-CNN sequences (both *p* < 0.05). Conclusions: In MR imaging of the knee, DL-CNN sequences showed significantly higher image quality and diagnostic confidence compared to standard PROPELLER sequences, while reducing acquisition time substantially. Both sequences perform comparably in the detection of knee-joint pathologies, while DL-CNN sequences are superior for evaluation of retropatellar cartilage lesions.

## 1. Introduction

Magnetic resonance imaging (MRI) is the preferred imaging modality for assessment of soft tissues in and around joints and is thus widely used in musculoskeletal imaging [1,2,3,4]. In MR imaging of the knee joint, fast spin echo (FSE) sequences are usually acquired in 2D or 3D as they allow precise visualization of internal and external structures of the knee joint, such as menisci, cruciate and collateral ligaments, and cartilage, but also soft tissues, like muscle and fat around the joint [5,6]. However, motion artifacts, caused by blood flow of the adjacent vessels or patient motion, can hamper image quality and may be an important limiting factor when reporting MRI scans of the knee.

To increase time efficiency of MR scanners while simultaneously reducing motion artifacts, recent efforts have been focusing on decreasing scan times. A significant reduction in scan time in musculoskeletal imaging may be achieved using parallel imaging, simultaneous multislice acquisition, compressed sensing-based sampling, and synthetic MRI techniques [7,8]. While these techniques provide a significant reduction in acquisition time and artifacts, they also usually result in a lower signal-to-noise ratio (SNR) and loss of image quality when compared to conventional acquisition techniques [7]. Alternatively, the PROPELLER method which relies on radial k-space image acquisition can be used for reduction of motion artifacts. [9,10,11]. In the PROPELLER acquisition technique, data are collected using parallel lines in circle around the k-space, which helps minimizing motion artifacts while increasing scan times [9,12]. This technique effectively reduces motion artifacts while increasing image quality, and therefore has been successfully used in imaging of the joints [9].

Application of deep learning-based convolutional neural networks (DL-CNN) reconstruction has been recently described as an effective method to accelerate MRI scan times, while reducing image noise and maintaining optimal image contrast [13,14,15]. The combination of conventional FSE sequences with DL-CNN has already been successfully implemented in imaging of the shoulder and knee joint [14,16].

The combination of a radial k-space acquisition technique with DL-CNN image reconstruction has the potential to simultaneously reduce motion artifacts and image noise in addition to substantial scan time reduction. However, this has not been yet investigated, especially for the imaging of the joints.

Therefore, the aim of this study was to assess the diagnostic performance of standard radial k-space (PROPELLER) MRI sequences and compare them with accelerated acquisitions using the PROPELLER technique combined with a deep learning-based convolutional neural network (DL-CNN) reconstruction for evaluation of the knee.

## 2. Materials and Methods

This prospective cohort study received ethical board approval, and written informed consent was obtained from every patient included.

### 2.1. Participants

Between October and December 2021, patients referred for an MRI of the knee joint were prospectively included in the study. Two image sets of standard and accelerated MR PROPELLER sequences were acquired in each patient in coronal, sagittal, and axial plane. In total, 35 patients were included in the study (Flow chart in Figure 1).

### 2.2. MRI Examination

The MRI examinations were performed using a 1.5 Tesla MRI scanner (SIGNA Artist, Waukesha, WI, USA) with a dedicated 16-channel knee coil.

MR protocols included coronal T1-weighted, axial proton density (PD) fat-saturated (FS) and sagittal PD sequences. Standard and accelerated sequences were acquired using the PROPELLER technique. Accelerated sequences were further reconstructed using DL-CNN reconstruction AIR^TM^ Recon DL^®^ (Waukesha, WI, USA) with a medium level of SNR improvement (chosen between low, medium, and high), thereafter referred to as DL-CNN sequences. Table 1 shows detailed MRI parameters of the standard and accelerated sequences.

The AIR^TM^ Recon DL pipeline incorporates a deep convolutional neural network that operates on raw, complex-valued imaging data to generate a clean output image. Specifically, the CNN is designed to allow users to adjust the level of image noise reduction, minimize truncation artifacts and enhance edge sharpness. Integration into the scanner’s native reconstruction pipeline is crucial to access the complete, high bit-depth data. The CNN consists of approximately 10,000 kernels, totaling 4.4 million trainable parameters. Being a convolutional network, it can accommodate MR images of various sizes. The CNN was trained using a supervised learning approach, using pairs of images representing nearly perfect and conventional MRI images. The near-perfect training data consisted of high-resolution images with minimal ringing and very low noise levels. The conventional training data were synthesized from the near-perfect images, employing established methods to create lower-resolution versions with increased truncation artifacts and higher noise levels [17].

Subsequently, all image datasets were forwarded to the Picture Archiving and Communication System (PACS) of our department (IMPAX 6; Agfa-Gevaert N.V., Mortsel, Belgium) for subsequent analysis.

### 2.3. Assessment of Image Quality and Diagnostic Confidence

As a part of the initial reading training, a set of diverse knee joint examinations, including both standard sequences and DL-CNN sequences unrelated to the study, were reviewed by two readers. The readers consisted of a board-certified radiologist with 6 years of experience and a radiology resident with 3 years of experience. Readers were blinded to any clinical information and to the sequence identifiers. Discrepancies were thoroughly discussed until agreement was achieved.

Then, the same two readers assessed MRI images of the knee joint of 35 patients included in the study. All image sets underwent a process of removing sequence identifiers (standard sequence vs. DL-CNN sequences) and were subsequently combined in a mixed order. The readers conducted the review of all images in a randomized sequence. Following the completion of the readings, the sequence type information was disclosed for the purpose of statistical analysis.

Image quality of standard and DL-CNN PROPELLER sequences for tibial, femoral and retropatellar bone and cartilage, tibiofibular joint, muscle (vastus medialis and lateralis muscle), anterior and posterior cruciate ligament, medial and lateral collateral ligament, medial and lateral meniscus, and subcutaneous tissue at the level of the medial femoral condyle was assessed using a four-point Likert scale. The scale ranged from 0 (poor) to 4 (perfect), with intermediate ratings of 1 (moderate) and 2 (good).

Additionally, the diagnostic confidence for the aforementioned structures was evaluated, along with the assessment of contour sharpness and homogeneity of fat saturation in the central and peripheral field-of-view (FOV). A four-point Likert scale (0-poor, 1-moderate, 2-good, 3-perfect) was utilized for this evaluation. The central FOV was defined at the level of the knee joint, while the peripheral FOV was identified as the most medial part of the vastus medialis muscle. Motion artifacts were evaluated independently for each image set.

### 2.4. Assessment of Pathological Findings

Evaluation of bone was assessed as (0) normal bone, or (1) minimal, (2) moderate and (3) advanced productive changes as usually associated with osteoarthritis. The assessment of cartilage involved categorizing it into different levels: (0) normal and homogeneous, (1) superficial inhomogeneities with normal cartilage contour, (2) partial thickness cartilage loss of less than 50% without subchondral edema, and (3) full-thickness cartilage loss with subchondral edema. Soft tissues were assessed as (0) normal, (1) muscle edema, (2) muscle fatty degeneration.

Anterior and posterior cruciate ligament was categorized as (0) normal, (1) mucoid degeneration with slightly hyperintense signal in PD fs sequences, (2) strain or partial-thickness tear with high signal intensity, swelling or thinning of the ligament or wavy course, and (3) full-thickness tear with complete lack of continuity as described by Ng et al. [18]. Medial and lateral collateral ligament were characterized as (0) normal, (1) tendinopathy when the ligament was thickened with slightly hyperintense signal in PD fs sequences, (2) partial thickness tear, and (3) full-thickness tear with complete lack of ligamentous continuity. Medial and lateral meniscus was evaluated as follows: (0) normal, (1) mucoid degeneration, and (2) torn. The knee joint was assessed as (0) normal, or (1) minimal, (2) moderate, or (3) marked joint effusion.

Each structure was assessed in all planes of the acquired MR images and sequences.

### 2.5. Signal-to-Noise and Contrast-to-Noise Ratio

For quantitative assessment of image quality, the signal-to-noise ratio (SNR) and contrast-to-noise ratio (*CNR*) were measured for both sequences. Regions of interest (ROIs) with an area of 5 mm^2^ were placed individually on each set to determine the signal intensity (*SI*) in the bone (specifically the distal femur), muscle (vastus medialis muscle), and subcutaneous fat at the level of the medial femoral condyle. The noise was quantified as the standard deviation (*SD*) of the *SI* in an ROI measured in extracorporeal air.

The *SNR* and *CNR* were calculated using the following formulas:$$SNR=\frac{SI}{SD(air)}$$
$$CNR(bone)=\frac{SI\left(bone\right)-SI(muscle)}{SD\left(air\right)}$$
$$CNR(fat)=\frac{SI\left(fat\right)-SI(muscle)}{SD\left(air\right)}$$

### 2.6. Statistical Analysis

A Shapiro–Wilk test was applied to assess the normal distribution of findings [19,20,21]. A Wilcoxon signed-rank test was used to compare the findings of image quality and diagnostic confidence between standard and DL-CNN sequences [22]. If a significant difference between sequences was noticed, a Bonferroni–Holm post hoc test for multiple comparison was additionally performed [23].

All identified pathologies in the evaluated structures were documented individually for each reader and dichotomized as either not pathological (scores 0 and 1) or pathological (scores 2 and 3).

The agreement between standard and DL-CNN sequences, as well as the inter-reader agreement (IRA) for assessing image quality and diagnostic confidence, were determined using the intraclass coefficient (ICC) [24]. ICC values below 0.5 were considered poor, values between 0.5 and 0.75 were categorized as moderate, values between 0.75 and 0.9 were considered good, and values above 0.9 were regarded as excellent indicators of reliability [25].

Cohen’s Kappa statistic was applied for assessment of the IRA and agreement between standard and DL-CNN sequences in evaluation of the pathological findings of the knee joint. Kappa values between 0.41–0.60 were considered moderate, between 0.61–0.80 substantial, and above 0.81 almost perfect agreement [26,27]. *p*-value < 0.05 was considered significant. To calculate an appropriate minimum sample size, an a priori power analysis was conducted with following parameters: Cohen’s effect size was determined as 0.8, α = 0.05 and Power (1 − β) = 0.95 [28]. A power analysis was performed using a G* Power software, v. 3.1.9.1 (Heinrich-Heine-Universitaet Duesseldorf, Dusesseldorf, Germany) [29]. Other statistical analyses were conducted using SPSS, v. 26.0 (IBM, Armonk, NY, USA).

## 3. Results

A minimum sample size of 24 was calculated in the power analysis for comparison of two matched samples. Finally, 35 patients between 18 and 73 years of age (mean: 47 years old SD: 16 years old; male *n* = 17, female *n* = 18) were included in the study. The mean acquisition time of standard sequences was 10 min 3 s, and of accelerated sequences 4 min 45 s.

### 3.1. The Quality of the Images and the Level of Diagnostic Confidence

The average image quality scores for bone assessment were 2.0 for standard sequences and 2.9 for DL-CNN sequences, of cartilage 1.9 and 2.9, of anterior cruciate ligament 1.8 and 2.6, of posterior cruciate ligament 2.2 and 2.9, of medial meniscus 1.9 and 2.9, and of lateral meniscus 1.9 and 2.8, respectively.

The mean overall diagnostic confidence in standard and DL-CNN sequences was 2.0 and 2.9, respectively.

Significant better results were observed in the average image quality and diagnostic confidence of all analyzed structures when utilizing DL-CNN sequences in comparison to the standard sequences (*p* < 0.05). No motion artifacts were detected in any of the image sets.

Overall, there was a moderate and good IRA for image quality in standard and DL-CNN sequences with ICC of 0.52 and 0.83. The inter-reader agreement for the assessment of diagnostic confidence in both standard and DL-CNN sequences showed moderate agreement, with ICC values of 0.54 and 0.55, respectively. There was a good agreement between standard and DL-CNN sequences for assessment of image quality with ICC of 0.77 and a moderate agreement for diagnostic confidence with ICC of 0.59.

For detailed information, please refer to Table 2 and Table 3. Figure 2 shows examples of both sequences.

### 3.2. Pathological Findings

Tibial and femoral bones were evaluated as normal (grades 0 and 1) in 16 and 18 patients, and in 19 and 17 patients as pathologic (grades 2 and 3) in standard and DL-CNN sequences, respectively. Tibial and femoral cartilage was evaluated as normal in 24 and 23 patients, and in 11 and 12 patients as pathologic (grades 2 and 3) in standard and DL-CNN sequences. ACL was assessed as normal in 30 patients in both standard and DL-CNN sequences and as pathologic in 5 patients. PCL was considered pathologic in one and two patients in standard and DL-CNN sequences, respectively. Medial meniscus was evaluated as pathologic in 10 and 9 patients in standard and DL-CNN sequences, while lateral meniscus in 5 and 4 patients.

In general, there was a moderate inter-reader agreement when assessing pathological findings in the knee joint using both standard and DL-CNN sequences, as indicated by Kappa values of 0.54 and 0.55, respectively. There was a substantial level of agreement between standard and DL-CNN sequences in the assessment of pathological findings of the knee joint with a κ-value of 0.8.

The summary of results of pathological findings of all analyzed structures in standard and DL-CNN sequences and IRAs is depicted in Table 4. Figure 3, Figure 4 and Figure 5 showcase various examples of pathological findings observed in both standard and DL-CNN sequences.

### 3.3. Signal-to-Noise and Contrast-to-Noise Ratios

The mean signal-to-noise ratio for bone, fat, and muscle was significantly higher in DL-CNN sequences compared to standard sequences (*p* < 0.05). Mean CNR was significantly higher in DL-CNN than in standard sequences (*p* < 0.05). For details refer to Figure 6.

## 4. Discussion

This study represents the first attempt to integrate PROPELLER MR acquisition technique with DL-CNN image reconstruction for knee joint imaging (Appendix A Appendix A).

Deep-learning based reconstruction algorithms in musculoskeletal imaging can be applied both to minimize image noise and to reduce the scan time. Recht et al. described that DL can be used to reconstruct fourfold accelerated images resulting in higher image quality than standard sequences. Moreover, the DL-accelerated sequences performed interchangeably with standard sequences for the detection of pathology of the knee [14]. Additionally, a super-resolution technique using deep learning and convolutional neural networks have been used for post-processing of the lower-resolution MR images of the knee joint and abdomen, resulting in superior image quality and diagnostic performance compared to the standard sequences [30,31,32,33]. As already shown in different studies, deep learning reconstructions can be also used for computed tomography (i.e., for planning of different orthopedic implants), as they serve as a robust method for dose reduction, while maintaining the image quality [34,35,36,37].

The PROPELLER acquisition technique is used when correction of motion and pulsatile flow artifacts is needed and has been described for imaging of the joints, head, and abdomen [9,10,11,12,38,39]. In 2011, Dietrich et al. reported that the PROPELLER technique in the MRI of the shoulder can be successfully used to reduce motion artifacts, while increasing image quality [9]. Similarly, application of radial k-space acquisition technique for imaging of the knee joint in the study of Lavdas et al. resulted in significant elimination of motion and pulsatile flow artifacts and thus higher image quality when compared to the conventional fast spin-echo (FSE) sequences [11] at the expense of increased scan time [9,12].

We sought to combine advantages of both approaches, i.e., artifact reduction by the motion insensitive PROPELLER technique with image noise and scan time reduction by using DL-CNN image reconstruction. The results confirmed a significant improvement in both image quality and diagnostic confidence when utilizing DL-CNN sequences in comparison to standard sequences, while reducing acquisition time from 10 min 3 s to 4 min 45 s. As expected, there were no motion artifacts in any of the image sets that would have hampered image analysis.

In our study, there was a significantly superior image quality of the DL-CNN sequences compared to the standard PROPELLER for imaging the knee joint, subsequently resulting in higher diagnostic confidence. Moreover, the IRA for image quality was better in DL-CNN than in standard sequences with ICC of 0.830 and 0.524, correspondingly.

A moderate level of IRA was observed for the evaluation of diagnostic confidence in both standard and DL-CNN sequences. We assume that a new imaging technique might have caused a different image impression to the radiologists who were experienced in reading conventional MRI sequences, resulting in different subjective perception of analyzed images.

The substantial agreement between standard and DL-CNN sequences for assessment of pathological findings of the knee joint is consistent with the fact that pathologies could be properly detected in both standard and DL-CNN sequences. The DL-CNN sequences however were significantly superior for the assessment of retropatellar cartilage, especially for evaluation of superficial cartilage defects and chondral delamination. Different methods for imaging and segmentation of knee cartilage have been defined including ultra-short echo time and T1_ρ_ sequences and deep convolutional neural networks; however, the utilization of the PROPELLER technique combined with DL-CNN reconstruction for knee cartilage imaging has not been described thus far [32,40,41,42]. Our study shows that a PROPELLER acquisition technique can be successfully applied for imaging of the knee joint in combination with DL-CNN sequences and enables superior depiction of superficial cartilage defects and cartilage delamination.

Our study has limitations. First, this is a small pilot study including only 35 patients that looks at different structures of the knee joint. Large sample sizes and a more focused study hypothesis may be warranted for future studies comparing standard to DL-CNN MR imaging. Second, all sequences were acquired using the PROPELLER technique and no conventional FSE sequences as part of most standard MR protocols were included. However, this study was specifically designed to address the lack of literature regarding the application of the PROPELLER technique in generating DL-CNN sequences [16,38,43]. Finally, there was no arthroscopic reference standard for the different joint pathologies detected by MRI. However, as described in previous studies, MRI is a reliable non-invasive method for evaluation of the knee joint with high correlation with arthroscopic findings [5,44].

In conclusion, MR imaging of the knee using accelerated PROPELLER sequences combined with DL-CNN reconstructions showed significantly higher image quality and diagnostic confidence compared to standard PROPELLER sequences, while reducing acquisition time substantially from 10 min 3 s to 4 min 45 s. Both sequences perform comparably in the detection of knee-joint pathologies, while DL-CNN images are superior for evaluation of retropatellar cartilage lesions.

These findings suggest that a combination of PROPELLER acquisition and the DL-CNN reconstruction technique may become the new reference standard for motion-artifact free, fast MR imaging of the knee with a superior depiction of superficial cartilage defects and cartilage delamination compared to a standard PROPELLER technique.

## Figures and Tables

**Figure 1 diagnostics-13-02438-f001:**
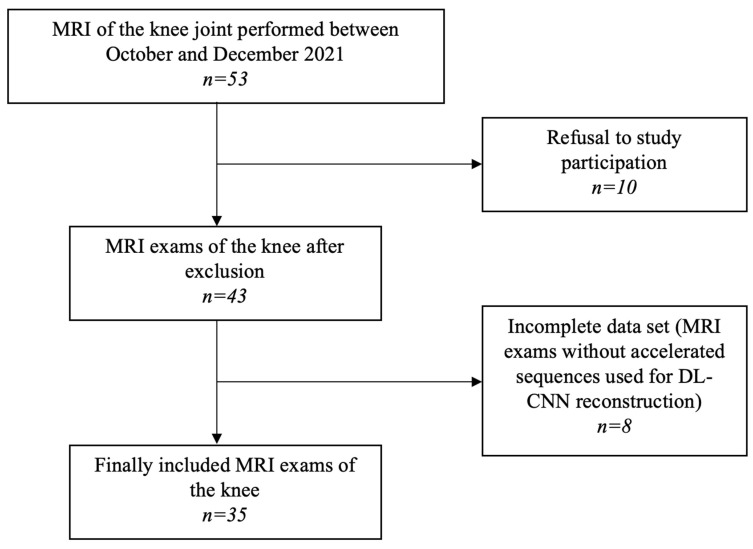
Flow diagram of patient selection. Data search in 3 different data bases (PubMed, Scopus and Web of Science).

**Figure 2 diagnostics-13-02438-f002:**
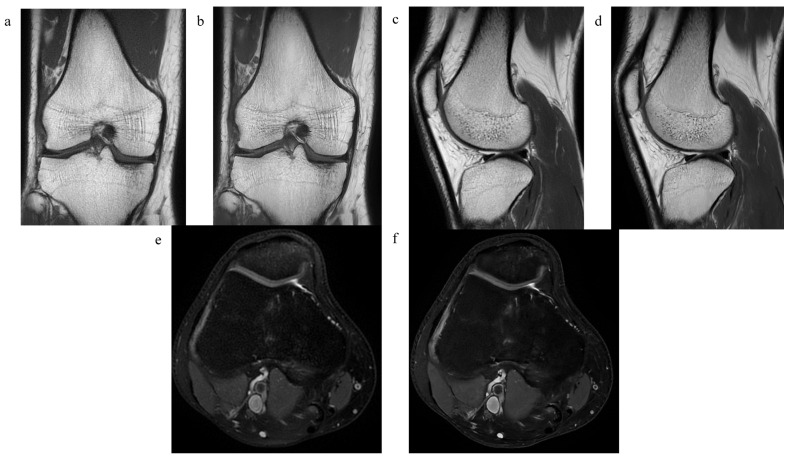
MR images of the right knee joint of a 38-year-old male without any symptoms. (**a**) Coronal T1-weighted (T1w) standard PROPELLER MR image, (**b**) coronal T1w DL-CNN MR image, (**c**) sagittal proton-density (PD) standard PROPELLER MR image, (**d**) sagittal PD DL-CNN MR image, (**e**) axial PD FS PROPELLER MR image, and (**f**) axial PD FS DL-CNN MR image.

**Figure 3 diagnostics-13-02438-f003:**
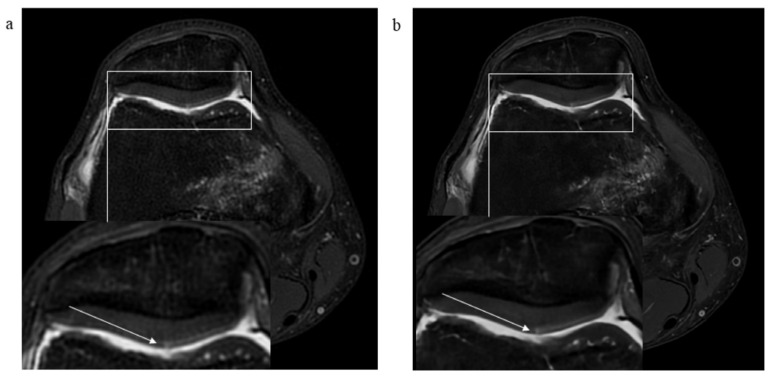
MR images of the right knee joint from a 44-year-old male experiencing knee pain following a direct trauma to the medial aspect of the knee. (**a**) Axial proton-density (PD) fat-saturated (FS) standard PROPELLER MR image, and (**b**) axial PD FS DL-CNN MR images show a superficial defect (arrow) of patellar cartilage with a horizontal course along lateral facet of the patella suggestive of a non-displaced delamination. Note a bone marrow edema of medial femoral condyle due to a direct trauma.

**Figure 4 diagnostics-13-02438-f004:**
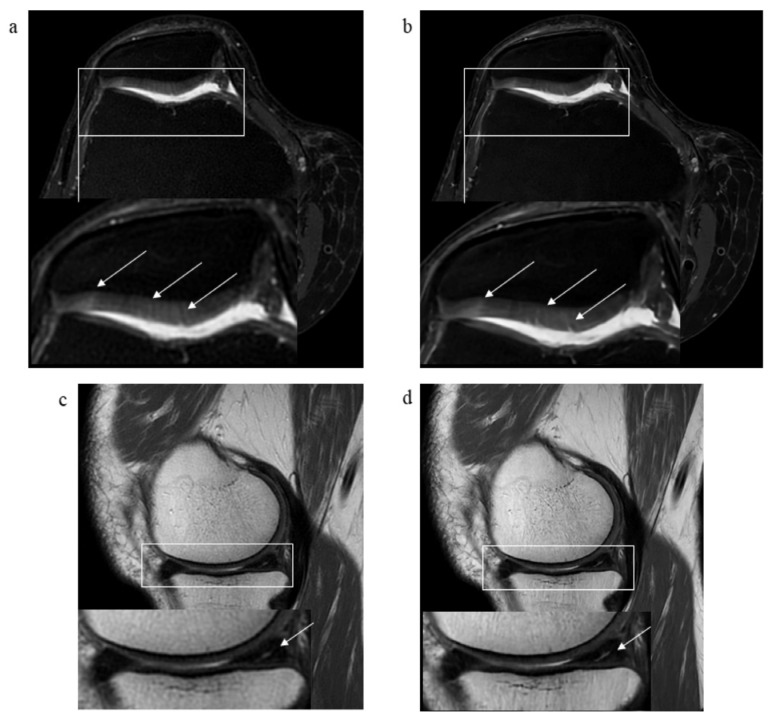
MR images of the right knee joint from a 63-year-old male presenting with pain and a clicking sound in the knee joint after a twisting injury. (**a**) Axial proton-density (PD) fat-saturated (FS) standard PROPELLER and (**b**) axial PD FS DL-CNN MR images show subtle superficial defects of patellar cartilage (arrows). (**c**) Sagittal PD standard PROPELLER and (**d**) sagittal PD DL-CNN MR images reveal a horizontal tear of posterior horn of the medial meniscus extending to the tibial surface (arrow).

**Figure 5 diagnostics-13-02438-f005:**
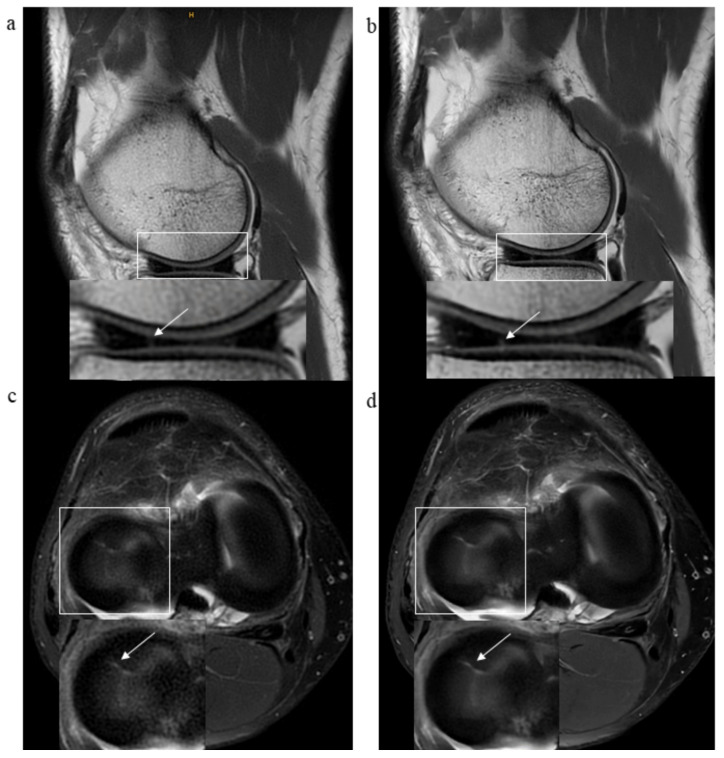
MR images of the right knee joint from a 51-year-old male suffering from chronic knee pain. (**a**) Sagittal proton density (PD) standard PROPELLER, (**b**) sagittal PD DL-CNN MR images and (**c**) axial PD FS PROPELLER, and (**d**) axial PD FS DL-CNN MR images show a radial tear of anterior horn of the lateral meniscus (arrow). Pathologies can be depicted in both sequences; however, DL-CNN sequences are less noisy, so pathological findings can be easier recognized.

**Figure 6 diagnostics-13-02438-f006:**
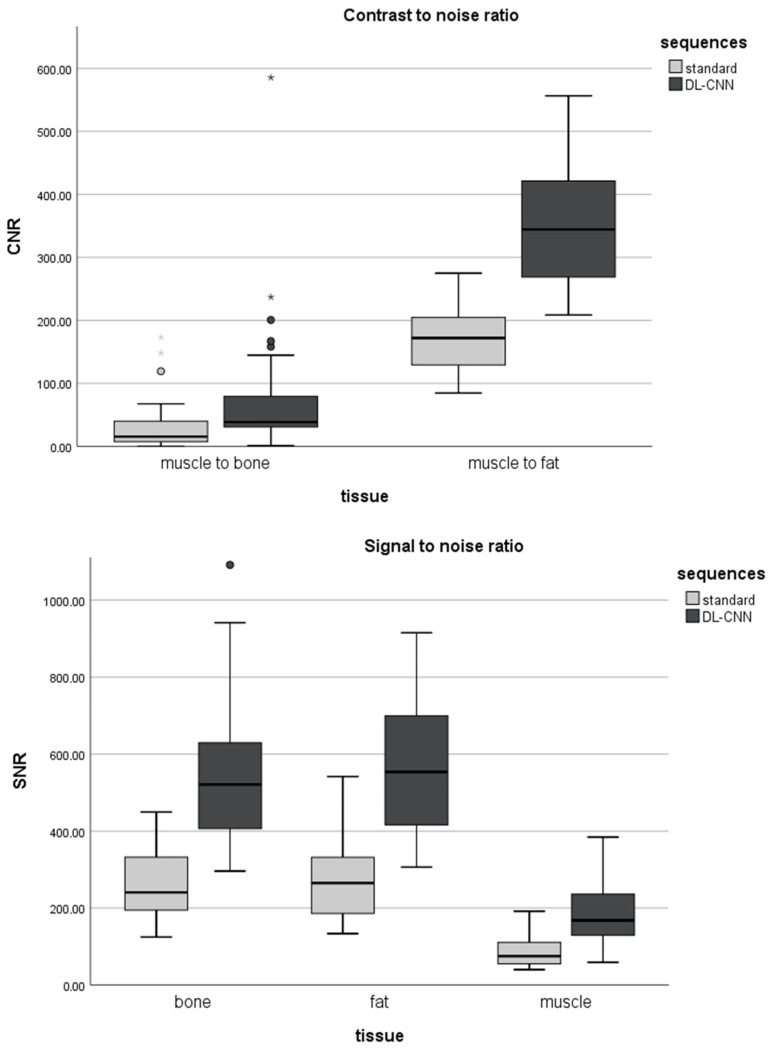
Signal-to-noise ratio (SNR) for bone, fat, and muscle (**top**) and contrast-to-noise ratios (CNR) (**bottom**) for standard and DL-CNN sequences. The mean signal-to-noise ratio for bone, fat, and muscle was significantly higher in DL-CNN sequences compared to standard sequences (*p* < 0.05). Mean CNR was significantly higher in DL-CNN than in standard sequences (*p* < 0.05).

**Table 1 diagnostics-13-02438-t001:** Standard sequences and DL-CNN PROPELLER sequences. ax—axial, cor—coronal, FS—fat saturated, PD—proton density, sag—sagittal oblique, SL—slice thickness, T1w—T1-weighted, T2w—T2-weighted, TE—echo time, TR—repetition time.

	Standard PROPELLER			DL-CNN Sequences		
	ax PD FS	sag PD	cor T1w	ax PD FS	sag PD	cor T1w FS
TE (ms)	49.9	44.2	12.8	46.7	42.9	42.2
TR (ms)	4649	2517	511	4421	2500	541
SL (mm)	3	3	3	3	3	3
Spacing between slices	4	4	4	4	4	4
Echo train length	15	14	7	15	14	7
Echo numbers	1	1	1	1	1	1
Matrix	300 × 300	320 × 320	340 × 340	320 × 320	360 × 360	360 × 360
Flip angle (°)	160	160	160	160	160	160
Receiver bandwidth (kHz)	162.7	195.3	195.3	195.3	244.1	244.1
Number of averages	3.0	2.8	2.6	1.6	1.5	1.6
Imaging frequency	63.8	63.8	63.8	63.8	63.8	63.8
Acquisition time (min:s)	04:09	02:46	03:08	02:00	01:09	01:36
Overall acquisition time (min:s)	10:03			04:45		

**Table 2 diagnostics-13-02438-t002:** The image quality of all evaluated structures in the knee joint was assessed using both standard and DL-CNN sequences. The results of the Wilcoxon-signed rank test for comparing the MR sequences are provided. The evaluation of bone quality included femoral, tibial, and retropatellar bones. Patellar cartilage was evaluated separately. Image quality was measured using four-point Likert scale (0-poor, 1-moderate, 2-good, 3-perfect).

	Standard PROPELLER (Mean ± SD)	DL-CNN (Mean ± SD)	Wilcoxon-Signed Rank Test (*p*-Value)
Bone	2.05 ± 0.57	2.90 ± 0.08	
Cartilage	1.95 ± 0.29	2.85 ± 0.22	
Retropatellar cartilage	1.94 ± 0.05	2.95 ± 0.50	
Muscle	2.22 ± 0.51	2.42 ± 0.16	
Anterior cruciate ligament	1.80 ± 0.05	2.60 ± 0.51	
Posterior cruciate ligament	2.24 ± 0.31	2.97 ± 0.20	
Medial collateral ligament	1.98 ± 0.02	2.91 ± 0.11	
Lateral collateral ligament	1.95 ± 0.02	2.88 ± 0.05	
Medial meniscus	1.91 ± 0.11	2.82 ± 0.30	
Lateral meniscus	2.04 ± 0.20	2.92 ± 0.86	
Contour sharpness in central FOV	1.95 ± 0.42	2.92 ± 0.02	
Contour sharpness in peripheral FOV	2.05 ± 0.40	2.85 ± 0.32	
Homogeneity of fat saturation in central FOV	2.32 ± 1.05	2.40 ± 0.65	
Homogeneity of fat saturation in peripheral FOV	1.84 ± 1.34	2.30 ± 0.71	all < 0.05

**Table 3 diagnostics-13-02438-t003:** Diagnostic confidence of all analyzed structures of the knee joint in standard and DL-CNN sequences together with results of Wilcoxon-signed rank test for MR sequence comparison. The knee structures were assessed as groups as follows: Bone and cartilage were assessed in the femoral, tibial, and patellar bone and retropatellar cartilage; muscles were evaluated in the medial and lateral vastus muscle; ligaments were assessed as the anterior and posterior cruciate ligament, and medial and lateral collateral ligaments. The assessment of diagnostic confidence was conducted utilizing a four-point Likert scale, with ratings ranging from 0 (poor) to 3 (perfect).

	Standard PROPELLER (Mean ± SD)	DL-CNN (Mean ± SD)	Wilcoxon Signed-Rank Test (*p*-Value)
Bone	2.07 ± 0.47	2.94 ± 0.23	
Cartilage	1.91 ± 0.40	2.85 ± 0.35	
Muscle	2.41 ± 0.55	2.92 ± 0.25	
Ligaments	2.14 ± 0.48	2.87 ± 0.33	
Medial and lateral meniscus	2.12 ± 0.52	2.89 ± 0.31	
Subcutaneous fat tissue	2.49 ± 0.55	2.90 ± 0.33	
Overall	2.04 ± 0.31	2.95 ± 0.20	all < 0.05

**Table 4 diagnostics-13-02438-t004:** The summary of pathologies observed in all analyzed structures using both standard and DL-CNN sequences, along with the corresponding inter-reader agreements, is presented. The assessment of inter-reader agreement was performed using Cohen’s Kappa statistic. Kappa values between 0.41–0.60 were considered moderate, between 0.61–0.80 substantial, and above 0.81 almost perfect agreement.

	Assessment of Pathologies		Inter-Reader Agreement	
	Standard PROPELLER (Mean ± SD)	DL-CNN (Mean ± SD)	Standard PROPELLER (κ-Value)	DL-CNN (κ-Value)
Bone	1.87 ± 0.89	1.80 ± 0.89	0.479	0.508
Cartilage	1.31 ± 0.46	1.34 ± 0.47	0.739	0.832
Patellar cartilage	1.37 ± 0.48	1.45 ± 0.50	0.756	0.770
Anterior cruciate ligament	1.12 ± 0.33	1.14 ± 0.35	0.624	0.717
Posterior cruciate ligament	1.02 ± 0.16	1.04 ± 0.20	0.653	0.729
Medial collateral ligament	1.12 ± 0.33	1.12 ± 0.33	0.873	0.873
Lateral collateral ligament	1.01 ± 0.11	1.01 ± 0.11	0.810	0.814
Medial meniscus	1.27 ± 0.44	1.24 ± 0.43	0.784	0.767
Lateral meniscus	1.14 ± 0.35	1.11 ± 0.32	0.710	0.720
Joint effusion	1.32 ± 0.47	1.31 ± 0.46	0.807	0.736

## Data Availability

Research data are to be found on the local computer of the University Hospital of Zurich and it is its property.

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
