# Peer review of "Deep Learning Convolutional Neural Network Reconstruction and Radial k-Space Acquisition MR Technique for Enhanced Detection of Retropatellar Cartilage Lesions of the Knee Joint"

_diagnostics, 2023, doi:10.3390/diagnostics13142438_

Round 1

Reviewer 1 Report

In MR imaging of the knee DL-CNN sequences showed significantly higher image quality and diagnostic confidence compared to standard PROPELLER sequences while reducing scan time substantially. Both sequences perform comparably in the de-tection of knee joint pathologies, while DL-CNN sequences are superior for evaluation of peripatellar cartilage lesions. Some remark given for this manuscript as follows.

1.      In the beginning of introduction section, the authors has state fast spin echo as FSE, why in discussion still state as fast spin echo? Evaluate and revise it.

2.      The present study using magnetic resonance imaging (MRI) for scanning, why not using computed tomography (CT)? Please give the rationalisation in the revised manuscript along with supporting reference as follows, doi: 10.3390/ma16093298

3.      The procedure of MR technique still not clear.

4.      The authors use DL-CNN, not CNN? Any explanation?

5.      The definition of “PROPELLER” needs more clearly highlight.

6.      The difference of PROPELLER and “standard” PROPELLER shout be explained.

7.      What is the current article novel? It has been extensively discussed in the past. Nothing truly novel in its current state. The absence of anything original makes the current study seem like a replication or a modified study. The introduction section should contain specifics about the writers' uniqueness. It is a significant reason to reject this study.

8.      In table 1 the authors state “sag- sagittal”, please recheck.

-

Author Response

Dear Reviewer

Thank you very much for possibility of proving a revision of our manuscript. We have profoundly revised the manuscript according to the Reviewer´s comments and changed the manuscript. Please find the revised manuscript and point-by-point answers to the Reviewers’ comments for your consideration.

Reviewer 1

In MR imaging of the knee DL-CNN sequences showed significantly higher image quality and diagnostic confidence compared to standard PROPELLER sequences while reducing scan time substantially. Both sequences perform comparably in the detection of knee joint pathologies, while DL-CNN sequences are superior for evaluation of peripatellar cartilage lesions. Some remark given for this manuscript as follows.

Thank you very much for your feedback. We have thoroughly revised our manuscript according to your comments. Please find the details in the point-to-point response below.

  1. In the beginning of introduction section, the authors have stated fast spin echo as FSE, why in discussion still state as fast spin echo? Evaluate and revise it.

Thank you for this suggestion. We have accordingly revised our manuscript and corrected this sentence as follows:

“Similarly, application of radial k-space acquisition technique for imaging of the knee joint in the study of Lavdas et al. resulted in significant elimination of motion and pulsatile flow artifacts and thus higher image quality when compared to the conventional fast spin-echo (FSE) sequences, at the expense of increased scan time. "

  1. The present study using magnetic resonance imaging (MRI) for scanning, why not using computed tomography (CT)? Please give the rationalisation in the revised manuscript along with supporting reference as follows, doi: 10.3390/ma16093298

Thank you very much for this question and your suggestion.

Magnetic resonance imaging is the preferred imaging modality for assessment of soft tissues in and around joints and is thus widely used in musculoskeletal imaging. This is also a first-line modality for assessment of the knee joint, as it allows precise visualization of internal and external structures of the knee joint, such as menisci, cruciate and collateral ligaments, cartilage, but also soft tissues, like muscle and fat around the joint. Therefore, we have chosen this established imaging modality for evaluation of the knee joint.

As already shown in different studies, deep learning reconstructions can be also used for computed tomography, especially as a robust method for dose reduction while maintaining the image quality.  

We have accordingly revised our manuscript and included the suggested reference as follows:

“As already shown in different studies, deep learning reconstructions can be also used for computed tomography (i.e., for planning of different orthopedic implants), as they serve as a robust method for dose reduction while maintaining the image quality (34, 35).

  1. The procedure of MR technique still not clear.

Thank you very much for this feedback. The applied deep-learning algorithm works on the following basis:

The AIRTM Recon DL pipeline includes a deep convolutional neural network that operates on raw, complex-valued imaging data to produce a clean output image. Specifically, the convolutional neural network is designed to provide a user tunable reduction in image noise, to reduce truncation artifacts, and to improve edge sharpness. Integration into the scanner’s native, inline reconstruction pipeline is critical as this provides access to raw, full bit-depth data. The convolutional neural network contains 4.4 million trainable parameters in approximately 10,000 kernels. It is a convolutional network, making it suitable for all MR relevant images sizes. The convolutional neural network was trained with a supervised learning approach using pairs of images representing near-perfect and conventional MRI images. The near-perfect training data consisted of high-resolution images with minimal ringing and very low noise levels. The conventional training data were synthesized from near-perfect images using established methods to create lower resolution versions with more truncation artifacts and with higher noise levels.

Due to this, application of deep learning-based convolutional neural networks such as above described AIRTM Recon DL reconstructions have been recently described as an effective method to accelerate MRI scan times, while reducing image noise and maintaining optimal image contrast.

  1. The authors use DL-CNN, not CNN? Any explanation?

Thank you for this remark.

Deep-learning reconstructions used in our study include a deep convolutional neural network that operates on raw, complex-valued imaging data to produce a clean output image. To emphasize the combination of deep-learning methods, which are derived from convolutional neural networks, we coined the term deep learning-based convolutional neural networks, thus “DL-CNN”.

  1. The definition of “PROPELLER” needs more clearly highlight.

Thank you very much for this suggestion. As required, we have thoroughly described the PROPELLER technique and now it reads as follows:

Radial k-space sampling by periodically rotated overlapping parallel lines with enhanced reconstruction known as PROPELLER technique can be used to reduce motion artifacts. In PROPELLER acquisition technique, data is collected in concentrical parallel lines rotating around the k-space, leading to correction of spatial variations and reduction of motion artifacts at the expense of increased scan time. This technique effectively reduces motion artifacts while increasing image quality, and therefore has been successfully used in imaging of the joints.”

  1. The difference of PROPELLER and “standard” PROPELLER should be explained.

We have addressed this point and added information on the PROPELLER acquisition technique and deep-learning reconstruction. Please find it below:

Combination of radial k-space acquisition technique with DL-CNN image reconstruction has the potential to simultaneously reduce motion artifacts and image noise in addition to substantial scan time reduction.

Therefore, the aim of this study was to assess diagnostic performance of standard radial k-space (PROPELLER) MRI sequences and compare with accelerated acquisitions using PROPELLER technique combined with a deep learning-based convolutional neural network (DL-CNN) reconstruction for evaluation of the knee joint (…).

MR protocols included a coronal T1-weighted, axial proton density (PD) fat-saturated (FS) and sagittal PD sequences. Standard and accelerated sequences were acquired using PROPELLER technique. Accelerated sequences were further reconstructed using DL-CNN reconstruction AIRTM Recon DL®

  1. What is the current article novel? It has been extensively discussed in the past. Nothing truly novel in its current state. The absence of anything original makes the current study seem like a replication or a modified study. The introduction section should contain specifics about the writers' uniqueness.

 We agree with the Reviewer that different studies have been published on deep-learning reconstructions and algorithms, as an effective method to accelerate MRI scan times, while reducing image noise and maintaining optimal image contrast. Moreover, the combination of conventional FSE sequences with DL-CNN has already been successfully implemented in imaging of the shoulder and knee joint.

However, we present the first study with combination of PROPELLER MR acquisition technique with a DL-CNN image reconstruction for imaging of the knee joint. We sought to combine advantages of both approaches, i.e., artifact reduction by motion insensitive PROPELLER technique with image noise and scan time reduction by using DL-CNN image reconstruction. This is the uniqueness of this study, in which accelerated PROPELLER sequences combined with DL-CNN reconstructions showed significantly higher image quality and diagnostic confidence compared to standard PROPELLER sequences while reducing scan time substantially. Therefore, these sequences and such approach may be of utmost importance in clinical routine and should be considered, especially in uncooperative patients, where motions artifacts and long scan times can significantly hamper image quality.

  1. In table 1 the authors state “sag- sagittal”, please recheck.

We have rechecked it.

The abbreviation “sag” stays for sagittal oblique MR images of the knee joint, that is usually used for imaging of injuries of the anterior cruciate ligament (ACL).

Reviewer 2 Report

Dear authors;

Thank you for the opportunity to deal with your respected work.

You clearly demonstrated the merit of DL-CNN reconstruction technique to improve MRI imaging of the musculoscelletal apparatus - especially in terms of diagnostic time.

Remarkably, the imaging quality was improved under the use of this new protocol while the diagnostic time was nearly halved.

Please display this important finding of significant time reduction more evidently in your results section. To me- this is your best finding.

Excellent work overall

KR

Clemens Schopper

Author Response

Dear Reviewer

Thank you very much for possibility of proving a revision of our manuscript. We have profoundly revised the manuscript according to the Reviewer´s comments and changed the manuscript. Please find the revised manuscript and point-by-point answers to the Reviewers’ comments for your consideration.

Reviewer 2

Dear authors;

Thank you for the opportunity to deal with your respected work.

You clearly demonstrated the merit of DL-CNN reconstruction technique to improve MRI imaging of the musculoscelletal apparatus - especially in terms of diagnostic time.

Remarkably, the imaging quality was improved under the use of this new protocol while the diagnostic time was nearly halved.

Please display this important finding of significant time reduction more evidently in your results section. To me- this is your best finding.

Excellent work overall

Thank you very much for your positive feedback, we really appreciate it. As suggested, we have emphasized the time reduction in our conclusion and now it reads as follows:

In conclusion, in MR imaging of the knee using accelerated PROPELLER sequences combined with DL-CNN reconstructions showed significantly higher image quality and diagnostic confidence compared to standard PROPELLER sequences while reducing scan time substantially from 10 minutes 3 seconds to 4 minutes 45 seconds. Both sequences perform comparably in the detection of knee joint pathologies, while DL-CNN images are superior for evaluation of retropatellar cartilage lesions.”

Round 2

Reviewer 1 Report

Appreciate to the authors in the previous revision, but some remarks needs to address as follows.

1.      The authors novel still not clear, needs to highlighted.

2.      Line 308, the authors uses the terms of “To the best of our knowledge..”, please proof it with literature searching from three main database, there are Scopus, PubMed, and Web of Science.

3.      Line 319-322, related the authors explanation, please incorporated additional literature to support the explanation of computed tomography in orthopaedic implant as follows: https://doi.org/10.1016/j.heliyon.2022.e12050

4.      Line 331, what is the authors meaning of “scan time”? It is ambiguous.

-

Author Response

Dear Reviewer

Thank you very much for possibility of providing a second revision of our manuscript. We have revised the manuscript as required and changed the manuscript according to your comments. Please find the revised manuscript and point-by-point answers to your comments below.

Reviewer 1

Appreciate to the authors in the previous revision, but some remarks needs to address as follows.

  1. The authors novel still not clear, needs to highlighted.

Thank you very much for this suggestion. We have changed the manuscript accordingly and now it reads as follows:

Combination of radial k-space acquisition technique with DL-CNN image reconstruction has the potential to simultaneously reduce motion artifacts and image noise in addition to substantial scan time reduction. However, this has not been yet investigated especially for the imaging of the joints.

Therefore, the aim of this study was to assess diagnostic performance of standard radial k-space (PROPELLER) MRI sequences and compare with accelerated acquisitions using PROPELLER technique combined with a deep learning-based convolutional neural network (DL-CNN) reconstruction for evaluation of knee.”

  1. Line 308, the authors uses the terms of “To the best of our knowledge..”, please proof it with literature searching from three main database, there are Scopus, PubMed, and Web of Science.

Thank you very much for this feedback. We have searched all databases, including Scopus, PubMed, and Web of Science did not find any studies on this topic. However, to avoid any confusion we changed this paragraph as follows:

This study represents the first attempt to integrate PROPELLER MR acquisition technique with DL-CNN image reconstruction for knee joint imaging.”

  1. Line 319-322, related the authors explanation, please incorporated additional literature to support the explanation of computed tomography in orthopaedic implant as follows: https://doi.org/10.1016/j.heliyon.2022.e12050

We have added this reference as suggested.

“As already shown in different studies, deep learning reconstructions can be also used for computed tomography (i.e. for planning of different orthopedic implants), as they serve as a robust method for dose reduction while maintaining the image quality (34-36).  “

  1. McLeavy CM, Chunara MH, Gravell RJ, Rauf A, Cushnie A, Staley Talbot C, et al. The future of CT: deep learning reconstruction. Clinical Radiology. 2021;76(6):407-15. doi: https://doi.org/10.1016/j.crad.2021.01.010.
  2. Salaha ZFM, Ammarullah MI, Abdullah NNAA, Aziz AUA, Gan H-S, Abdullah AH, et al. Biomechanical Effects of the Porous Structure of Gyroid and Voronoi Hip Implants: A Finite Element Analysis Using an Experimentally Validated Model. Materials. 2023;16(9):3298. PubMed PMID: doi:10.3390/ma16093298.
  3. Jamari J, Ammarullah MI, Santoso G, Sugiharto S, Supriyono T, Permana MS, et al. Adopted walking condition for computational simulation approach on bearing of hip joint prosthesis: review over the past 30 years. Heliyon. 2022;8(12). doi: 10.1016/j.heliyon.2022.e12050.

  1. Line 331, what is the authors meaning of “scan time”? It is ambiguous.

Thank you very much for this question. For the clarity, we have changed the possibly confusing “scan time” to “acquisition time” and accordingly corrected the manuscript.

Round 3

Reviewer 1 Report

Some clarification needs to addressed by the authors, please find it below.

1.      Regarding revision round 2, comments number 2. If the author’s state “We have searched all databases, including Scopus, PubMed, and Web of Science did not find any studies on this topic”, please provide the searching results in form of graph and table to indicate “The best of our knowledge”. For example, searching from 2000-2023 with article type from Scopus, PubMed, and Web of Science using keywords “knee”, “deep learning”, and others in year to year shows in Table/Figure x. Refining the results with…….. and manual investigation, several similar literatures found present in Table/Figure x. Then the authors do not find it which shows the groundbreaking novel in the present study. If the authors can not proof it, better remove the statement “To the best of our knowledge”.

2.      Please explain potential further study performing computational simulation/in silico in medical application. It brings several advantages, such as lower cost and faster results compared to clinical/in vivo and laboratory/in vitro. Provide this information along with relevant reference as follows: https://doi.org/10.3390/biomedicines11030951

-

Author Response

Dear Reviewer

Thank you very much for possibility of providing a third revision of our manuscript. We have revised the manuscript as required and changed the manuscript according to your comments. Please find the revised manuscript and point-by-point answers to your comments below.

  1. Regarding revision round 2, comments number 2. If the author’s state “We have searched all databases, including Scopus, PubMed, and Web of Science did not find any studies on this topic”, please provide the searching results in form of graph and table to indicate “The best of our knowledge”. For example, searching from 2000-2023 with article type from Scopus, PubMed, and Web of Science using keywords “knee”, “deep learning”, and others in year to year shows in Table/Figure x. Refining the results with…….. and manual investigation, several similar literatures found present in Table/Figure x. Then the authors do not find it which shows the groundbreaking novel in the present study. If the authors can not proof it, better remove the statement “To the best of our knowledge”.

Thank you very much for this suggestion. As required, we have added a Table with a detailed data search in Pubmed, Scopus and Web of Science, showing results in each databank for different keywords. Please find it below.

Database

Pubmed

Search Keywords

"deep learning AND propeller AND MRI"

Results

Title

Authors

Journal

Year

1

Deep Learning and Imaging for the Orthopaedic Surgeon: How Machines "Read" Radiographs.

Hill BG, Krogue JD, Jevsevar DS, Schilling PL. 

J Bone Joint Surg Am.

2022

2

Comparison of deep learning-based reconstruction of PROPELLER Shoulder MRI with conventional reconstruction

Hahn S, Yi J, Lee HJ, Lee Y, Lee J, Wang X, Fung M

Skeletal Radiol.

2023

3

Impact of Deep Learning Reconstruction Combined With a Sharpening Filter on Single-Shot Fast Spin-Echo T2-Weighted Magnetic Resonance Imaging of the Uterus.

Tsuboyama T, Onishi H, Nakamoto A, Ogawa K, Koyama Y, Tarewaki H, Tomiyama N.

Invest Radiol.

2022

4

Scientific Advances and Technical Innovations in Musculoskeletal Radiology.

Fritz J, Runge VM.

Invest Radiol.

2023

5

Application of deep learning-based image reconstruction in MR imaging of the shoulder joint to improve image quality and reduce scan time.

Kaniewska M, Deininger-Czermak E, Getzmann JM, Wang X, Lohezic M, Guggenberger R.

Eur Radiol.

2023

6

A Feasibility Study on Deep Learning Reconstruction to Improve Image Quality With PROPELLER Acquisition in the Setting of T2-Weighted Gynecologic Pelvic Magnetic Resonance Imaging.

Saleh M, Virarkar M, Javadi S, Mathew M, Vulasala SSR, Son JB, Sun J, Bayram E, Wang X, Ma J, Szklaruk J, Bhosale P.

J Comput Assist Tomogr. 

2023

Database

Pubmed

Search Keywords

"deep learning AND propeller AND MRI AND knee"

Results

"No results were found."

Database

Scopus

Search Keywords

"deep learning AND propeller AND MRI"

Results

Title

Authors

Journal

Details

1

Comparison of deep learning-based reconstruction of PROPELLER Shoulder MRI with conventional reconstruction

Hahn, S., Yi, J.,Lee, H.-J., .Wang, X., Fung, M.

Skeletal Radiology

52(8), pp. 1545–1555

2

Application of deep learning–based image reconstruction in MR imaging of the shoulder joint to improve image quality and reduce scan time

Kaniewska, M., Deininger-Czermak, E., Getzmann, J.M., .Lohezic, M., Guggenberger, R. 

European Radiology

33(3), pp. 1513–1525

3

Breast Diffusion MRI Acquisition and Processing Techniques: The GE Healthcare Perspective

Shimakawa, A., Bayram, E.

Diffusion MRI of the Breast

pp. 251–25

4

The Reconstruction Method Using Compressed Sensing and Convolutional Neural Network for PROPELLER MRI in Head

Matsumoto, Y., Hori, K., Tadano, K., .Endo, Y., Hashimoto, T. 

2021 IEEE Nuclear Science Symposium and Medical Imaging Conference Record

 NSS/MIC 2021 and 28th International Symposium on Room-Temperature Semiconductor Detectors, RTSD 2022,

Database

Scopus

Search Keywords

"deep learning AND propeller AND MRI AND knee"

Results

"No documents matching your keywords were found."

Database

Web of Science

Search Keywords

"deep learning AND propeller AND MRI"

Results

Title

Authors

Journal

Details

1

Comparison of deep learning-based reconstruction of PROPELLER Shoulder MRI with conventional reconstruction

Hahn, S; Yi, J; (...); Fung, M

SKELETAL RADIOLOGY

52 (8), pp.1545-1555

2

Application of deep learning-based image reconstruction in MR imaging of the shoulder joint to improve image quality and reduce scan time

Kaniewska, M; Deininger-Czermak, E; (...); Guggenberger, R

EUROPEAN RADIOLOGY

 33 (3) , pp.1513-1525

3

Impact of Deep Learning Reconstruction Combined With a Sharpening Filter on Single-Shot Fast Spin-Echo T2-Weighted Magnetic Resonance Imaging of the Uterus

Tsuboyama, T; Onishi, H; (...); Tomiyama, N

INVESTIGATIVE RADIOLOGY

57 (6) , pp.379-386

4

Dual-domain self-supervised learning for accelerated non-Cartesian MRI reconstruction

Zhou, B; Schlemper, J; (...); Sofka, M

MEDICAL IMAGE ANALYSIS

81

5

Comprehensive Clinical Evaluation of a Deep Learning-Accelerated, Single-Breath-Hold Abdominal HASTE at 1.5 T and 3 T

Herrmann, J; Wessling, D; (...); Othman, AE

ACADEMIC RADIOLOGY

30 (1) , pp.93-102

6

Pediatric brain extraction from T2-weighted MR images using 3D dual frame U-net and human connectome database

Kim, D; Chae, JH and Han, Y

INTERNATIONAL JOURNAL OF IMAGING SYSTEMS AND TECHNOLOGY

29 (4) , pp.476-482

7

Stochastic optimization of three-dimensional non-Cartesian sampling trajectory

Wang, GH; Nielsen, JF; (...); Noll, DC

Magn Reson Med

Apr 2023 (Early Access)

8

Motion artifact reduction for magnetic resonance imaging with deep learning and k-space analysis

Cui, L; Song, Y; (...); Yang, G

PLOS ONE

18 (1)

Database

Web of Science

Search Keywords

"deep learning AND propeller AND MRI AND knee"

Results

"No records were found to match your filter"

  1. Please explain potential further study performing computational simulation/in silico in medical application. It brings several advantages, such as lower cost and faster results compared to clinical/in vivo and laboratory/in vitro. Provide this information along with relevant reference as follows: https://doi.org/10.3390/biomedicines11030951

We have added this reference as suggested.

  1. McLeavy CM, Chunara MH, Gravell RJ, Rauf A, Cushnie A, Staley Talbot C, et al. The future of CT: deep learning reconstruction. Clinical Radiology. 2021;76(6):407-15. doi: https://doi.org/10.1016/j.crad.2021.01.010.

  1. Salaha ZFM, Ammarullah MI, Abdullah NNAA, Aziz AUA, Gan H-S, Abdullah AH, et al. Biomechanical Effects of the Porous Structure of Gyroid and Voronoi Hip Implants: A Finite Element Analysis Using an Experimentally Validated Model. Materials. 2023;16(9):3298. PubMed PMID: doi:10.3390/ma16093298.

  1. Jamari J, Ammarullah MI, Santoso G, Sugiharto S, Supriyono T, Permana MS, et al. Adopted walking condition for computational simulation approach on bearing of hip joint prosthesis: review over the past 30 years. Heliyon. 2022;8(12). doi: 10.1016/j.heliyon.2022.e12050.

  1. Ammarullah MI, Hartono R, Supriyono T, Santoso G, Sugiharto S, Permana MS. Polycrystalline Diamond as a Potential Material for the Hard-on-Hard Bearing of Total Hip Prosthesis: Von Mises Stress Analysis. Biomedicines. 2023;11(3):951. PubMed PMID: doi:10.3390/biomedicines11030951.
